# A novel system to culture human intestinal organoids under physiological oxygen content to study microbial-host interaction

Tatiana Y. Fofanova[1�], Umesh C. Karandikar[1☍], Jennifer M. Auchtung[1,2], Reid L. Wilson[3,4], Antonio J. Valentin[5], Robert A. Britton[1], K. Jane Grande-Allen[3], Mary K. Estes[5,6], Kristi Hoffman[1], Sashirekha Ramani[5], Christopher J. Stewart[1,7]*, Joseph F. Petrosino[1,5]*

1 Alkek Centre for Metagenomics and Microbiome Research, Department of Molecular Virology and Microbiology, Baylor College of Medicine, Houston, TX, United States of America, 2 Department of Food Science and Technology, University of Nebraska-Lincoln, Lincoln, NE, United States of America, 3 Department of Bioengineering, Rice University, Houston, TX, United States of America, 4 Medical Scientist Training Program, Baylor College of Medicine, Houston, TX, United States of America, 5 Department of Molecular Virology and Microbiology, Baylor College of Medicine, Houston, TX, United States of America, 6 Department of Medicine, Baylor College of Medicine, Houston, TX, United States of America, 7 Translational and Clinical Research Institute, Newcastle University, Newcastle, United Kingdom

☍ These authors contributed equally to this work.
* christopher.stewart@newcastle.ac.uk (CJS); jpetrosi@bcm.edu (JFP)

**Data Availability Statement:** All relevant data are within the manuscript and its Supporting Information files.

## Abstract

Mechanistic investigation of host-microbe interactions in the human gut are hindered by difficulty of co-culturing microbes with intestinal epithelial cells. On one hand the gut bacteria are a mix of facultative, aerotolerant or obligate anaerobes, while the intestinal epithelium requires oxygen for growth and function. Thus, a coculture system that can recreate these contrasting oxygen requirements is critical step towards our understanding microbial-host interactions in the human gut. Here, we demonstrate *Intestinal Organoid Physoxic Coculture* (IOPC) system, a simple and cost-effective method for coculturing anaerobic intestinal bacteria with human intestinal organoids (HIOs). Using commensal anaerobes with varying degrees of oxygen tolerance, such as nano-aerobe *Bacteroides thetaiotaomicron* and strict anaerobe *Blautia* sp., we demonstrate that IOPC can successfully support 24–48 hours HIO-microbe coculture. The IOPC recapitulates the contrasting oxygen conditions across the intestinal epithelium seen *in vivo*. The IOPC cultured HIOs showed increased barrier integrity, and induced expression of immunomodulatory genes. A transcriptomic analysis suggests that HIOs from different donors show differences in the magnitude of their response to coculture with anaerobic bacteria. Thus, the IOPC system provides a robust coculture setup for investigating host-microbe interactions in complex, patient-derived intestinal tissues, that can facilitate the study of mechanisms underlying the role of the microbiome in health and disease.

**Funding:** U19-AI116497 Mary K Estes P30 DK-56338 H. EL-Sarag T32GM088129, and F30 DK-108541 Wilson Reid The funders had no role in study design, data collection and analysis, decision to publish, or preparation of the manuscript.

**Competing interests:** The authors have declared that no competing interests exist.

## Introduction

The human gastrointestinal tract is a site for digestion, immune system regulation, drug and nutrient absorption, and host-environment interactions [1]. The microbiome can affect each of these processes by influencing metabolism, immune balance, and therapeutic outcomes [2]. Adult intestinal stem cell-derived cultures, established from biopsies or surgical samples and termed human intestinal organoids (HIOs), present a unique opportunity for studying host-microbe interactions at the intestinal epithelium [3–5]. These patient-derived epithelial cultures can be differentiated into all the major cell types of the intestinal epithelium and demonstrate physiologic activity consistent with their region of isolation [6–9]. In addition, as HIOs retain the genetic signatures of the individual from whom they were derived [10–12], they are uniquely advantageous models for investigating donor-specific interactions between hosts and microbes [13]. Further, HIOs are a significant advancement towards implementation of the three Rs principles (Replacement, Reduction and Refinement) in conducting humane animal research [14].

Modeling host-commensal interactions under dynamic, biologically relevant pericellular oxygen conditions is critical to understanding host-microbe interactions in the human gut. However, laboratory studies to mechanistically investigate these interactions present multiple challenges. First, the intestinal epithelium is oxygen dependent, while many gut bacteria are obligate anaerobes. Thus, the intestinal epithelium marks the transition between the anaerobic intestinal lumen on its apical side and the physiological oxygen levels in the lamina propria on its basal side. Recreating this contrasting oxygen conditions across the single-cell thick epithelial layer poses a significant design challenge [15]. Second, achieving physiological levels of oxygen is critical to study the intestinal epithelium. The intestinal epithelium is oxygenated by the counter-current blood flow in the vasculature of the crypt-villus axis. As the intestinal epithelium resides in close proximity to the lamina propria with diverse cell types including circulating immune cells [16], it is kept in a state of constant, low-grade hypoxia known as "physoxia". When compared to the physoxia experienced by the intestinal epithelium, cells maintained under standard cell culture conditions experience higher levels of oxygen. It has been shown that gene-expression and epithelial phenotypes are influenced by oxygen levels especially when comparing cells grown in ambient incubator conditions to cells grown under physoxic conditions [17, 18]. These results suggest that ambient oxygen levels can potentially lead to atypical phenotypes under routine cell culture conditions.

Recent advances in organoid based models to address these challenges have led to coculture systems with a range of benefits and limitations [19]. In the last few years, two transwell based coculture systems using HIOs have been described to study host-microbe interactions. The first system called Intestinal Hemi-Anaerobic Coculture (iHACS), has a hypoxic apical chamber and a normoxic basal chamber [20]. While this set up is simple, it is not amenable to frequent sampling of the contents of apical chamber. The second culture system called Gut-MIcrobiome (GuMI) physiome platform described by Zhang et al. employs a custom 3D printed platform; this sophisticated platform allows for continuous sampling of both the apical and basal media [21]. However, the 3D printed cassette can accommodate co-culturing of a maximum of 8 transwells per set up, limiting the number of conditions that can studied in a single experiment. In addition, the pumps and custom 3D printed parts needed to maintain the flow make the overall setup too specialized for rapid adoption. Besides these two transwell based systems, 'organ on chip' microfluidic based specialty devices that employ flow to remove the toxic byproducts and replenish the depleting nutrients have been described [22]. However, these devices require extensive technical expertise, are labor-intensive, and expensive, which creates a barrier for routine use in research labs. Thus, there is a critical need for a cost-

effective, easy to assemble system to model host-microbiome interactions under physiologically relevant oxygen conditions. To address this gap, we developed the Intestinal Organoid Physoxic Culture (IOPC) system, which supports coculturing of intestinal anaerobes while allowing precise control over oxygen content.

## Results

### Assembly and validation of IOPC

In the intestine, the apical side of epithelial cells is exposed to anaerobic conditions in the lumen while the basal side receives dissolved oxygen diffusing from the blood vessels (**Fig 1A**). IOPC mimics this environment by exposing the HIO monolayer to anaerobic conditions on the apical side, while the basal side receives the dissolved diffusing from the growth medium. The oxygen in the growth media is replenished by a low oxygen gas mixture through a gas permeable membrane (arrowheads, **Fig 1B**). Unless mentioned otherwise, the low oxygen gas mixture in the IOPC consists of 5% $CO_2$, 5.6% $O_2$, and balance $N_2$. A detailed protocol for setting up the HIO monolayers in the IOPC is provided in the supplementary material. In brief, the IOPC set up (**Fig 1B**) uses HIO monolayers grown on transwells (**S1A Fig in S2 File**) in a 24-well plate with a gas-permeable bottom. The gas permeable bottom allows the media in the basal compartment to be saturated with the physoxic levels of $O_2$ from the gas mixture (**Fig 1B**). The plate is housed in an air-tight cassette with an inlet and an outlet to circulate the gas mixture (**Fig 1B and 1C**) without it leaking into the anaerobic chamber (**Fig 1D**). To confirm that IOPC supports robust cultivation of HIOs and validate the expected oxygen conditions, we analyzed the cell survival, dissolved oxygen content, and the gross morphology of a jejunal (Je) HIOs grown in IOPC and compared it with same line grown in standard culture conditions (henceforth referred to as IC- Incubator Control). After 24 hours of culture, there was no significant difference in viability (**Fig 1E**) between IOPC Vs IC grown Je HIOs. Hematoxylin and eosin (H&E) staining showed normal cellular morphology and polarity, while Alcian Blue staining confirmed the presence of mucus (**Fig 1F**) for the Je monolayers cultured in the IOPC set up.

To support survival of intestinal obligate anaerobic bacteria, any residual oxygen from the apical compartment needs to be eliminated. We theorized that in the IOPC set up, the HIO cells would scavenge residual oxygen and thereby create and maintain the anaerobic conditions on the apical side. To test this hypothesis and validate the contrasting oxygen conditions in the IOPC set up, we measured the dissolved oxygen in the media from the apical and basal compartments of the transwell using a clark-type microelectrode at 24 hours (**Fig 1G**). In the apical media, the oxygen concentration was below the limit of detection, while the oxygenation of the basolateral compartment reflected the $O_2$ content of the gas mix (5.6% or 10.2% oxygen). In contrast the transwell without the monolayer showed 3–4% $O_2$ in the apical media (**S1B Fig in S2 File**). Furthermore, we calculated that the $O_2$ levels in the apical compartment reached equilibrium by 2 hours (**S1C Fig in S2 File**). This time point was used to introduce intestinal anaerobes to the apical compartments in all the subsequent experiments.

### Physoxia-associated phenotypes seen in IOPC

It has been previously reported that 5.6% $O_2$ is sufficient to promote physoxia-associated phenotypes without substantial induction of the hypoxia marker HIF-1α [17]. Therefore, we investigated if IOPC cultured HIOs show these physoxia phenotypes such as elevated barrier integrity and immunomodulatory effects on the epithelial inflammatory response [23–25]. We measured trans-epithelial electrical resistance (TEER) as a proxy for epithelial barrier integrity in the IOPC grown Je HIOs and analyzed changes in gene expression using targeted, RT2

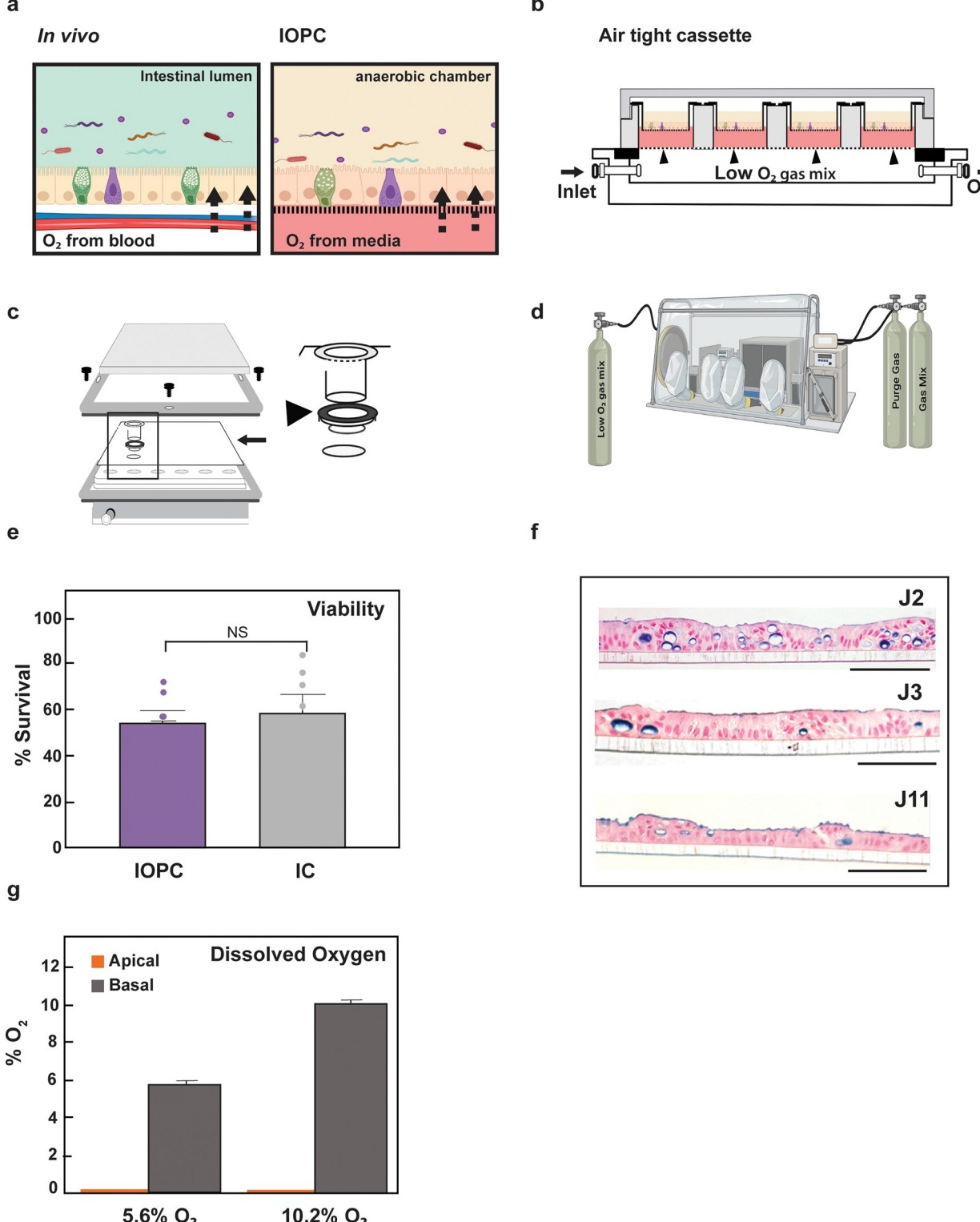

**Fig 1. Validation of the Intestinal Organoid Physoxic Coculture (IOPC).** (a) Schematic indicating the contrasting oxygen conditions across the intestinal epithelial cells *in vivo* and the HIO monolayer in IOPC. (b-d) Schematic details and overview of the IOPC set up. (b) Airtight cassette housing the 24-well plate with a gas-permeable bottom, and transwells with the Je HIO monolayers. The bottom compartment of the cassette is flooded with low O2 gas mix via the inlet. The gas mix exits via the anaerobic chamber via a tube connected to the outlet. Media is saturated with low levels of oxygen by diffusion across the gas permeable bottom indicated by the black arrowheads. (c) Assembly of the airtight cassette with a magnified

view corresponding to the inset (black rectangle). The transwells are fitted into a gasket (black arrowhead) and sealed in place using a double-sided adhesive tape (black arrow) on a 24-well plate. The 24 well plate is fixed into an airtight cassette with plastic screws to hold it in place. (d) A standard anaerobic cultivation chamber houses the airtight IOPC assembly with different gas tanks needed to recreate the physoxic oxygen conditions for the IOPC, the low oxygen gas mix is connected to the inlet (shown in panel b) of the airtight cassette via a tube through a port hole in the anaerobic chamber. (e) Cell viability of HIO monolayers cultured under IOPC (low O2 gas mix) compared to IC as measured by Trypan Blue dye exclusion (N = 7/group), NS-not significant. (f) H&E staining of the HIO monolayers showing the apico-basal polarity characteristic of jejunal epithelium. Alcian Blue staining was used to mark the mucus production. Scale bar, 100 microns. (g) Dissolved oxygen concentration in the apical and basolateral compartments of the IOPC system when supplied with low O2 gas mix containing different oxygen levels at 5.6% or 10.2%.

Profiler PCR Arrays for barrier integrity and immunomodulatory pathways. The TEER was significantly increased in Je HIOs cultured in IOPC system relative to ICs (**Fig 2A**). This change in TEER was confirmed in Je HIOs derived from multiple donors (J2, J3, J8, and J11) and Caco2 cells (**S2 Fig in S2 File**). Using the RT2 Profiler PCR Arrays, we compared the expression of 168 genes associated with epithelial barrier integrity, innate immunity, and anti-microbial response in IOPC and IC. We filtered our results to identify the genes that met the criteria for the cut-off (see methods). Of the 107 genes that met these criteria, 32 were significantly upregulated, while 3 genes were significantly downregulated in IOPC grown HIOs relative to IC (**Fig 2B**). Notably, 21 of the 32 upregulated genes across all three Je lines are known to be involved in epithelial barrier function via their known or predicted roles (**Fig 2C**). These include *INADL* and *PARD3* which are known to play a role in maintaining tight junctions, and *MPP6* and *CBR3* that are important for epithelial polarity, thus corroborating the TEER phenotype. IOPC cultured HIO lines further upregulated 12 genes that are known to play a role in anti-microbial response: for example, *TLR4*, *TLR6*, and *MYD88*. The three genes significantly downregulated in response to IOPC were *CXCL1* (*IL1*), *CXCL8* (*IL8*) and *TLR2* (**Fig 2E**) that are associated with inflammation. Heatmap analysis of the RT2 profiler array further suggested overall changes in gene expression of the different HIO lines in the IOPC system (**S3 Fig in S2 File**). Interestingly, these heatmaps also show line-specific changes to the gene expression in HIO lines.

Gene Ontology analysis highlighted several significantly activated pathways, including nitric oxide biosynthesis, nuclear factor kappa B (NFkB) activation, pattern recognition receptor (PRR) signaling, interleukin-6 (IL6) production, positive regulation of cell communication, inflammatory response activation, and maintenance of cell polarity (**S1 Table**). Upregulation of nitric oxide biosynthesis, one of the top 10 pathways, acts as a secondary validation that the changes observed are, in fact, a response to physiological hypoxia. Under higher oxygen conditions, the expression of the oxygen-sensitive transcription factor hypoxia-inducible factor-1alpha (HIF-1α) level is regulated by hydroxylation by prolyl hydroxylases (PHDs). We surmised that during reduced $O_2$ levels, endogenous nitric oxide disables PHDs, allowing HIF-1α to accumulate and activate its downstream gene targets. As expected, expression of HIF-1α was elevated in HIOs in IOPC (**Fig 2D**).

## IOPC-cultured HIOs support the coculture of intestinal anaerobes

Having confirmed that the IOPC can maintain anaerobic conditions on the apical side of the epithelial monolayer, we tested the ability of this system to support culturing of commensal bacteria. Members of the phyla Bacteroidetes and Firmicutes are among the most abundant bacteria in the human gut [26, 27]. We chose to focus on *Bacteroides thetaiotaomicron* and *Blautia* sp. as representatives of Bacteroidetes and Firmicutes respectively. *B. thetaiotaomicron* is a Gram-negative, acetate-producing nanoanaerobe [28], while *Blautia* sp. is a Gram-positive, lactate- and acetate-producing obligate anaerobe, an important member of the *Lachnospiraceae* family which is part of core gut microbiome member [29].

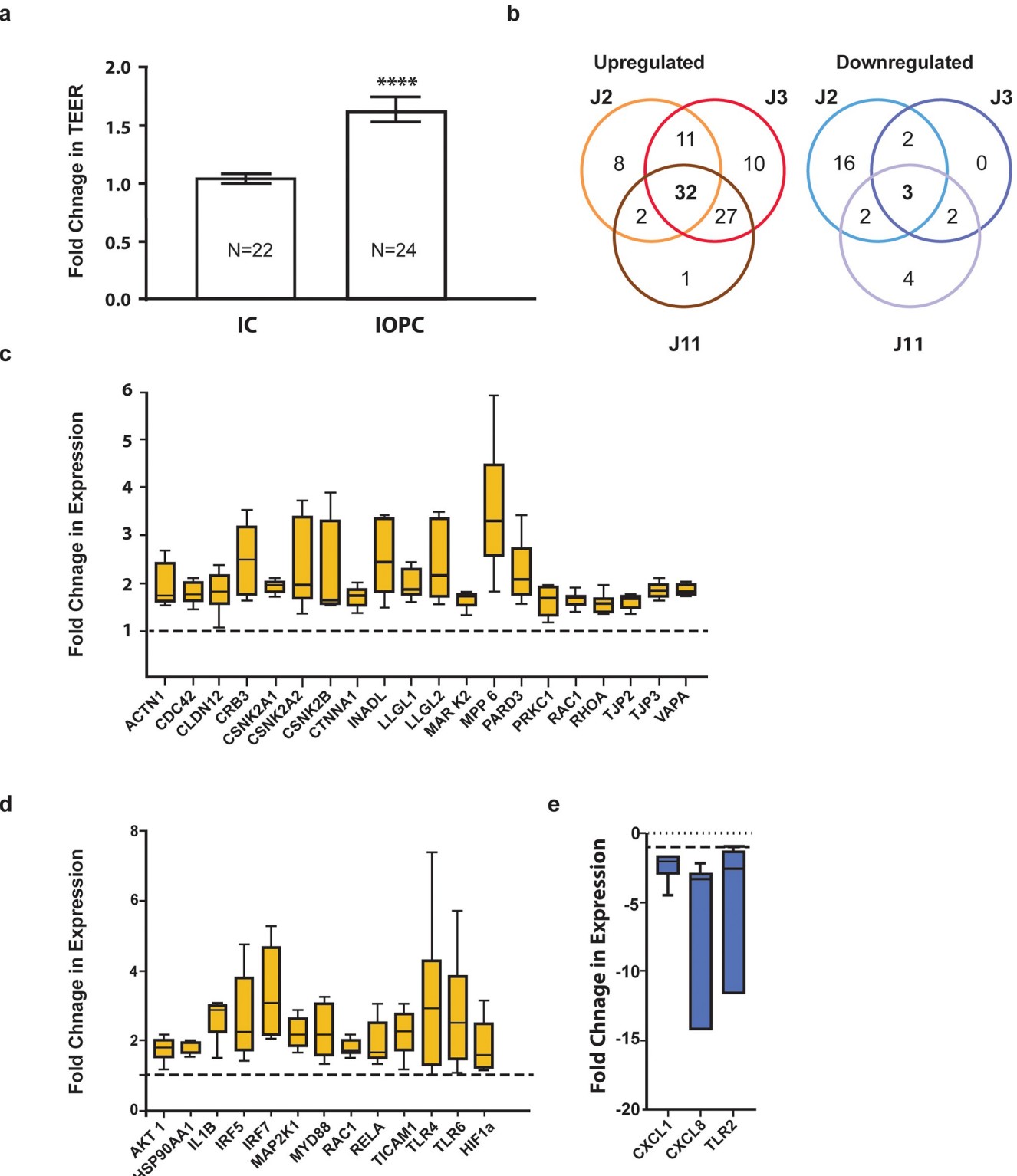

**Fig 2. Physoxic conditions in the IOPC alters trans-epithelial electrical resistance (TEER) and gene expression of intestinal epithelium.**
Characterization of the Je HIO monolayer under physoxic conditions in the IOPC system (a) HIO monolayers in IOPC show an increase in TEER when compared to IC. (b-e) Venn diagram and fold changes in gene expression across the three Je HIO lines–J2, J3 and J11 in the IOPC set up. (b) Venn diagram showing altered gene expression in HIOs cultured in IPOC compared to the IC. Genes that were consistently upregulated in all three HIOs include those involved in (c) barrier integrity, and (d) anti-microbial responses. (e) Genes that were consistently downregulated in the three Je HIO lines are involved in innate immune and inflammatory pathways. Dotted line indicates the baseline gene normalized expression in the IC cultured HIOs.

Following 2 hours of equilibration to achieve the physoxic conditions, approximately $10^{5-6}$ colony forming units (CFUs) of bacteria were inoculated into the apical compartment of the IOPC system (see **S4 Fig** in **S2 File** for the details). The IOPC system supported the survival of both *B. thetaiotaomicron* and *Blautia* sp. (black lines in **Fig 3A and 3B**). In the presence of the HIO monolayer, *B. thetaiotaomicron* showed a significant increase in CFUs over the 24 hr period, while in the absence of the HIO monolayer the CFUs remained relatively static. In contrast, the oxygen-sensitive *Blautia sp* showed some increase in CFUs measured at 8 hr. in the presence of the HIO monolayer while a drastic decline in the CFUs was seen in the absence of the HIO monolayer. These observations suggest that the growth of *B. thetaiotaomicron* and the survival of *Blautia sp*. was dependent on the presence of the HIO monolayer. The growth of *B. thetaiotaomicron* and presence of *Blautia sp*. was further confirmed using fluorescent in-situ hybridization for the 16S rRNA gene (**Fig 3C**). Interestingly, *B. thetaiotaomicron* is in closer contact with the epithelial surface at 8 hours and is subsequently detached from the epithelium by 24 hours.

We further tested the role of the microbes on epithelial barrier integrity by measuring the TEER. As expected, the J2, J3 and J11 HIOs showed enchanced TEER in IOPC as compared to ICs (**Fig 3D**). Cocutluring either *B. thetaiotaomicron* or *Blautia* sp. reduced the TEER of J2 and J11 lines as compared to the respective IOPC controls. However, the J3 line showed no discernable effect on TEER when cocultured with *B. thetaiotaomicron* as compared to its IOPC control. Further, the J3 showed a drastic drop in TEER in the presence of *Blautia sp*. (**Fig 3D**). This donor-specific effect of the presence of commensal microbes highlights the value of pairing organoids with the IOPC culture system and the need for evaluating host-microbe interactions in HIOs from multiple donors.

## Intestinal anaerobes induce transcriptomic changes to jejunal epithelium

Microbiota are known to induce changes in the intestinal epithelium, such as elevated barrier function and modulation of immune response [30]. Therefore, we evaluated whether the expression of genes regulating these functions was altered in IOPC HIOs in the presence of an intestinal microbe. We used the above mentioned RT2 profiler array targeting human tight junctions (TJ) and human antibacterial response in HIOs cocultured in the presence of *B. thetaiotaomicron*. A total of 18 genes were significantly upregulated across all three HIO lines when cocultured with *B. thetaiotaomicron* (**Fig 4A**) as compared to the IC controls. Eleven of those 18 genes were also upregulated in the IOPC controls while seven genes were upregulated exclusively in the presence of *B. thetaiotaomicron* (**Fig 4B**). Most of the upregulated genes affected different aspects of barrier integrity, including cytoskeleton regulation (*SPTAN1*, *SMURF1*) and junction maintenance (*ESAM*, *SYMPK*, *F11R*). On the other hand, 13 genes were significantly downregulated in HIOs cocultured with *B. thetaiotaomicron* coculture as compared to the IOPC controls. Three of the 13 genes were also downregulated in the IOPC controls while 10 genes were downregulated exclusively in response to *B. thetaiotaomicron* (**Fig 4C**). The intestinal epithelial genes that were downregulated exclusively in the presence of *B. thetaiotaomicron* included inflammatory response (*ICAM2*, *IL18*, *NFKB1*, *IL2*) and induction of apoptosis genes (*CASP1*, *PYCARD*) were strongly downregulated during coculture, suggesting an immunomodulatory role for *B. thetaiotaomicron* in the gut (**Fig 4D**). Interestingly some of the few genes involved in barrier integrity (*CLDN1/3*, *TJP2*) also showed reduction in gene expression. A heat map of the RT2 analysis suggests an overall reduced expression of these genes in HIOs cocultured with *B. thetaiotaomicron* as compared to HIO only controls for all three lines (**S5 Fig** in **S2 File**). Many of the genes significantly upregulated in IOPC were mitigated to IC levels of expression following *B. thetaiotaomicron* coculture (**S6 Fig** in **S2 File**).

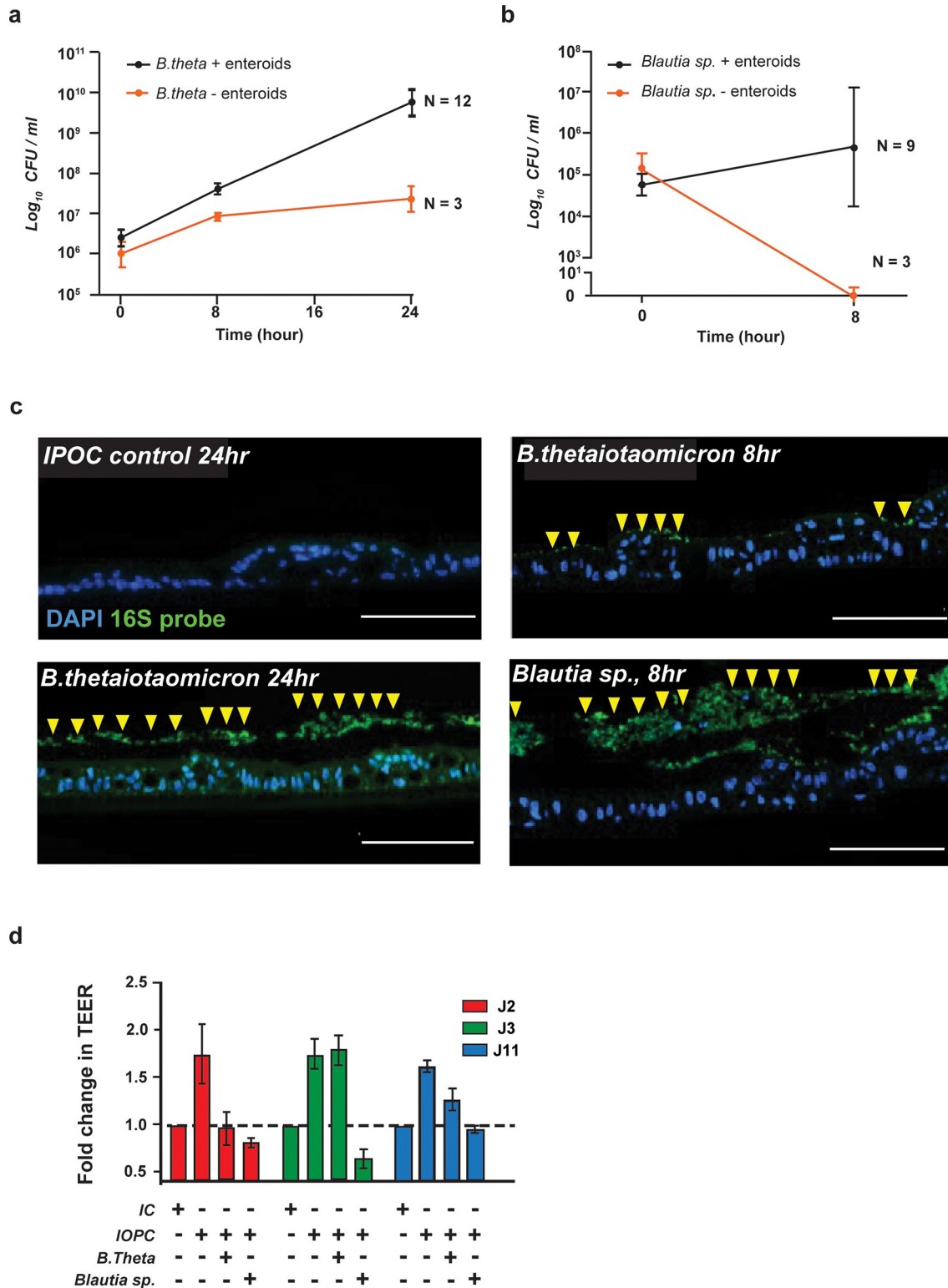

**Fig 3. The IOPC system supports coculture of HIO monolayers with anaerobic bacteria for up to 24 hours.** Characterization of the intestinal bacteria cocultured with Je HIOs as measured by bacterial viability (a & b) immunofluorescence (c) and the effect of intestinal bacteria on the barrier integrity of the Je HIOs (d). The Je HIOs grown in IOPC system supports the growth of (a) nanoanaerobe *B. thetaiotaomicron* (abbreviated *B. theta*) for 24 hours and the growth of obligately anaerobic (b)*Blautia sp.* for 8 hours in the presence of HIOs as indicated by the black lines. In contrast, *B. thetaiotaomicron* and *Blautia* are unable to grow in the

IOPC in the absence of HIOs as indicated by the orange line. (c) A Fluorescent In-Situ Hybridization probe against the 16S rRNA gene shows that the bacteria are localized to the apical compartment (yellow arrowheads), DAPI stained nuclei (blue) of the epithelial cells indicate the position of the HIO monolayer. Scale bar, 100 microns (d) Fold change in the TEER in HIO- microbe coculture in the IOPC. The dotted lines indicate the baseline TEER of the HIO lines grown in IC (n = 7 for IC, IOPC, and *B. thetaiotaomicron*-treated groups; n = 3 for *Blautia* treated groups).

Gene Ontology analysis of the activated genes suggested enrichment of several important pathways, including regulation of cell proliferation, inflammatory response, activation of immune response, cytokine production, NFkB activation, and response to a molecule of bacterial origin (S2 Table). These findings indicate that much of the changes to gene expression in the coculture are due to the physoxic culture conditions in the IOPC culture system. Overlap

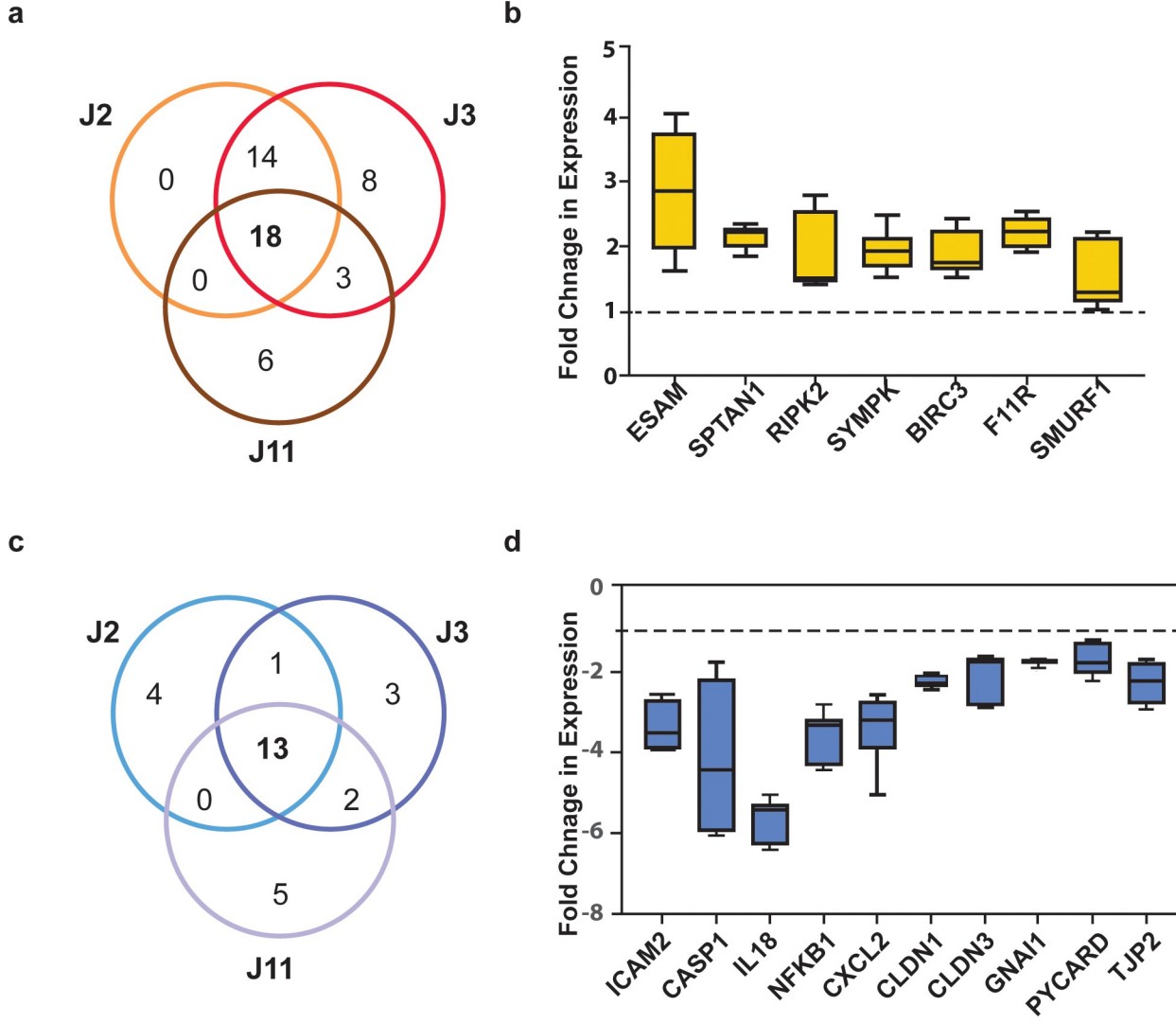

**Fig 4. Differential gene expression in Je HIOs is due to bacterial coculture.** Je HIOs cocultured with *B. thetaiotaomicron* showed differential regulation of gene expression, with sets of genes that were either consistently upregulated (a & b) or downregulated (c & d) across three HIO lines (J2, J3 and J11) in the IOPC setup. (a) Venn diagram showing 18 genes consistently upregulated across the Je HIO lines during coculture. (b) Fold change in expression of seven genes upregulated exclusively due to the presence of *B. thetaiotaomicron*. (c) Venn diagram showing 13 genes consistently downregulated across all three Je HIO lines during coculture. (d) Fold change in expression of the 10 genes downregulated in Je HIOs exclusively due to the presence of *B. thetaiotaomicron*. The dotted line shows the baseline gene expression in IOPC cultures without the microbes.

in gene expression profiles between jejunal HIO lines suggested that coculture may induce some common pathways in the intestinal epithelium.

## Discussion

We introduce IOPC, a simple organoid–microbiome coculture system that uses off the shelf parts, in this proof-of-concept report. IOPC successfully recapitulates the contrasting oxygen conditions across the intestinal epithelium *in vivo*, while supporting the growth and viability of the intestinal epithelium and the intestinal bacteria. Physiological oxygen concentrations in the intestinal mucosa ranges from 1–11%; yet most tissue culture experimentation is done at ~18% oxygen, making the standard tissue culture conditions hyperoxic. In addition, the tissue culture format influences oxygen content making a comparison across formats potentially misleading (12,14). For instance, when grown on membrane inserts or at the base of standard gas-permeable plates, the $O_2$ levels of Caco2 cultures can be as high as 16.5%. However, when seeded as monolayers on the base of a tissue culture grade plastic plate, the oxygen content of Caco2 cultures can fall as low as ~4% [31]. Based on our oxygen consumption models, the operating oxygen concentration for IOPC is approximately 2% - 10%. These values closely model oxygen concentrations found *in vivo*, which can range from 1% in inflamed tissues and tumors to 11% during meal digestion [32]. We demonstrate that Je HIOs cultured in the IOPC system exhibit several hallmarks of physoxia, including increased barrier integrity, upregulation of anti-microbial response genes involved in NFkB pathway, and reduced expression of pro-inflammatory genes such as IL-8 (**S1 Table**). Using *B.thetaiotaomicron* and *Blautia sp*., we have demonstrated the ability of IOPC to successfully sustain short-term coculture (i.e., up to 24 hours) of intestinal microbes with HIO monolayers. IOPC supports nano-anaerobe *B. thetaiotaomicron* growth and supports survival of the oxygen sensitive *Blautia sp*. The targeted analysis of transcriptional changes presented in this study indicate a significant impact of *B. thetaiotaomicron* on the barrier function and the innate epithelial immune response. Transcriptional changes in genes involved in barrier function and antibacterial function in the intestinal epithelial cells shown here are in line with a previous reported by Hill et al., in an iPSC-derived 3D organoid based coculture model [33].

Many host-microbe coculture systems have been described, and each provides a range of benefits and challenges. One of the earliest 'proof of concept' study used 3D organoids and employed injection of E.coli to demonstrate the use of organoid based models [33]. Most of the studies since then have focused on using organoid monolayers to study host-microbe interactions. Microfluidics-based systems offer flow, known to stimulate differentiation and physiologically relevant phenotypes [34] but may be limited by scalability and cost. Further, the highly technical nature of these systems means the use of these systems is restricted to specialist labs. Since we deposited this manuscript as a pre-print in bioRxiv circa 2019, two independent groups have published transwell-based systems to coculture intestinal microbes with HIO monolayers [20, 35]. Both these systems have unique strengths and limitations; for example, Sasaki et al. [20] describe a system that is relatively simple but requires inserting plugs that are not widely available, and the basal compartment is not physoxic. On the other hand, Zhang et al. describe a transwell system that supports longer-term coculture but needs a 3D printed manifold, an elaborate pump system to circulate the media while allowing only six samples per manifold [21]. Conversely, the IOPC system presented here is constructed from common laboratory and off-the-shelf materials that are readily available at relatively low costs. IOPC centers around existing infrastructure, such as anaerobic chambers that can be found in labs that routinely study anaerobic bacteria, with little to no modification. IOPC based HIO-microbe coculture permits direct interaction between human and microbial cells while allowing for

temporal sampling or replenishment of the apical media if desired by the user, which would allow for longer term coculture in the future.

Despite its value and utility, the IOPC as used in this study has several limitations. First, because the system is static, the HIO viability may be compromised by bacterial overgrowth and/ or changes in media pH over time. However, this limitation could be overcome through use of lower starting CFUs of bacteria, the use of slower growing bacteria, or manual replenishment of the media. Second, owing to the necessity to create an airtight seal between the basolateral and apical compartments, it is not possible to continually monitor the TEER for experiments aimed at evaluating changes in barrier integrity over time. However, TEER measurements can be readily obtained at the start and end of the experiment. Finally, an inherent limitation of HIOs and related systems derived from cell lines is the lack of circulating immune cells in the model, although methods exist to include these in Transwell monolayers [36]. Ongoing work to improve the IOPC will allow this integration of microbial-epithelial-immune cells into a single robust system in the future. Furthermore, we predict that IOPC can also aid in pharmacological studies to evaluate the absorption and transport of oral medications across the intestinal epithelium under physiological oxygen conditions in the presence of intestinal microbes.

## Conclusion

It is paramount that basic and therapeutic investigation of host-microbe interactions be conducted under oxygen conditions that are physiologically appropriate to the intestinal location and disease state in question. The IOPC system achieves this goal through use of commonly available, affordable materials, and requires minimal technical expertise for assembly and operation. It recapitulates the contrasting oxygen conditions seen *in vivo*, induces expression of physiological hypoxia-associated phenotypes, and sustains host-anaerobe interactions for at least 24 hours. Importantly, our results demonstrate that HIO monolayers cultured in IOPC system enhance barrier-integrity and anti-microbial responses in a donor specific manner. Based on this proof-of-concept report, we posit that further refinement of IOPC can significantly enhance the utility of patient specific HIOs to identify personalized list of beneficial microbes in a precision medicine approach [10–12] or intervention such as FMT.

## Materials and methods

### Human jejunal HIOs

Three different types of media were used to wash (complete medium without growth factors, CMGF-), proliferate (complete medium with growth factors, CMGF+), and differentiate/maintain (differentiation medium) HIOs, initially produced from human intestinal biopsies and cultured as detailed by Saxena *et al.* (2016). Three-dimensional cultures of jejunal HIOs were generated, cultured, and seeded as monolayers on 6.5 mm transwell inserts (COSTAR 3470). For monolayer formation, these dense 3D HIO cultures were prepared as single-cell suspensions using successive trypsin/EDTA incubation steps. Proliferating cultures were re-suspended in growth media (approximately $5x10^5$ cells / 100μL media) and seeded onto a diluted (1:40 in cold PBS) Matrigel/collagen coating film on the Transwell [9]. This process allows HIO monolayer formation. The growth medium was replaced with a differentiation medium after 1–2 days, once trans-epithelial electrical resistance approached 300 Ω, indicating monolayer confluence. Monolayers were differentiated for 4 days before performing the coculture experiments.

Specific HIOs used in each experiment are referred to by the intestinal segment of origin (J–Jejunum) followed by a number indicating donor origin (e.g., J2, J3, J8, J11 each originated from jejunum tissue from 4 different donors). Consent for the original donor tissue was obtained under IRB protocols H-13793 and H-31910 approved by the Baylor College of

Medicine Institutional Review Board. Established lines were used in all experiments when between 10–15 passages in culture.

## List of parts for the Intestinal Organoid Physoxic Culture

Coculture Chamber: BDEV035, Coy Laboratory Products

Gas permeable plates:

Double sided tape—D969PK Silicone/Acrylic Differential Tape by Specialty Tapes Manufacturing

Rubber gasket, Silicone grease, Blood gas (5% $CO_2$ / 5.6% or 10.2% $O_2$ / Balance $N_2$)

## HIO-anaerobe coculture system

HIOs were seeded as monolayers on 6.5 mm transwell inserts (COSTAR 3470). These were placed into modified gaskets that were sealed in place using double-sided adhesive tape on a 24-well gas-permeable plate. The entire apparatus was kept in an anaerobic chamber (90% $N_2$/ 5% $H_2$/5% $CO_2$) to allow growth of anaerobic bacteria on the apical surface. Gas (5% $CO_2$/balance $N_2$) containing oxygen (5.6% or 10.2%) was pumped from an external tank through the base of the plate to supply oxygen to the basolateral side of the monolayer. This constitutes a simple, cost-effective method for co-culturing obligate anaerobic bacteria with human, intestinal HIO monolayers under variable oxygen conditions. A detailed description of system design, setup, and operation can be found in the supplementary materials provided (**S1 File**).

## Bacterial strains, preparation, and quantification

Two commensal species, *Bacteroides thetaiotaomicron* (nano anaerobe) and *Blautia* sp. (obligate anaerobe), cultivated from a healthy human microbiome were used in this study. These microbes were selected to represent the two most abundant phyla in the human gut. We utilized the continuous-flow mini-bioreactor array system described in Auchtung *et al.* (2015) to ensure that the bacterial cultures remained consistent between experiments. Continuous-flow culture models allow for studies to be performed during an extended time period under conditions where pH, nutrient availability, and washout of waste products and dead cells can be well controlled. Notably, the IOPC system does not require mini-bioreactor arrays for operation and bacteria can be prepared according to user preference. Bioreactor medium was prepared as described in Auchtung *et al.* (2015) and inoculated with our two bacterial isolates. After inoculation of the bioreactor, bacteria were allowed to equilibrate for 16–18 h prior to the initiation of flow at 1.875 ml/h (8-h retention time).

Prior to coculture, approximately 0.2mL of bacterial culture was removed, serially diluted, and plated on bioreactor media agar plates (*B.theta*) or GM-17 agar plates (*Blautia* sp.). These were cultured under anaerobic conditions for 24–48 hours to determine viability and concentration (CFUs/mL). HIO monolayers were inoculated with either *B.theta* or *Blautia* sp. at $1x10^6$ CFU/mL or $1x10^5$ CFU/mL, respectively. *Blautia sp.* was inoculated at a lower concentration because of its considerably fast doubling time. At each experimentally defined timepoint, a 5μL sample was taken from the apical compartment of the Transwell, serially diluted, and spot plated in duplicate on the above described agar plates under anaerobic conditions to determine viability and concentration (CFUs/mL).

## Determination of oxygen concentration

Oxygen was measured with a 500 <u>u</u>M diameter Clark-type microelectrode (Unisense A/S) that was calibrated according to manufacturer's recommendation under anoxic and atmospheric

oxygen conditions. Measurements were taken at a consistent and defined position within the Transwell with the use of a micromanipulator to ensure precision.

## Simulation of oxygen transport in anaerobic model

A finite element simulation of oxygen transport in the IOPC was constructed in COMSOL Multiphysics 5.3. Two-dimensional, radially symmetrical models of the gas-permeable, anaerobic Transwell system were drawn to scale, and a very fine tetrahedral mesh was used to construct a wireframe. Dimensions of the 24 well plate and Transwell insert were obtained from the manufacturer's specifications. To simulate oxygen transport in the IOPC with 5.6% $O_2$, the concentration of $O_2$ at the top and bottom of the well were defined to be 0.164% and 5.677%, respectively, as measured empirically by Clark electrode. Diffusion coefficients for oxygen were assumed to be 1.78 x $10^{-6}$ $m^2$ $s^{-1}$ in the membrane and 3.0 x $10^{-9}$ $m^2$ $s^{-1}$ in the media. The semi-permeable Transwell membrane was modeled as a thin diffusion barrier with a diffusion coefficient equal to the diffusivity of the media multiplied by the porosity of the membrane (0.19).

Oxygen consumption by the HIO monolayer was modeled using Michaelis-Menten kinetics:

$$R_c = R_{max} \left( \frac{c}{c + C_{MM,O_2}} \right) \cdot \delta(c > C_{cr})$$

$R_c$ is the rate of consumption, $R_{max}$ is the maximum oxygen consumption rate, $c$ is the concentration of oxygen, $C_{MM,O_2}$ is the Michaelis-Menten constant equal to the oxygen concentration at which oxygen consumption is one half of the maximal rate, and $\delta$ is a step-down function that stops oxygen consumption when its concentration drops below the threshold that can sustain survival in long-term cultures ($C_{cr}$). Literature values were used for $C_{MM,O_2}$, 1.0 x $10^{-3}$ mol $m^{-3}$ and $C_{cr}$, 1.0 x $10^{-4}$ mol $m^{-3}$.

$R_{max}$ was determined experimentally by culturing HIO monolayers on collagen type I gels (Rat Tail Collagen Type I, Corning) in an oxygen-sensing well plate (OP96C, PreSens Precision Sensing GmbH) for 5 days in a standard tissue culture incubator. In this setting, $R_c$ approximates $R_{max}$ due to the high $O_2$ concentration. The equilibrium $O_2$ concentration of the monolayers was measured, and $R_{max}$ was calculated using a finite element model of the well plate.

## Measurement of trans-epithelial resistance

The barrier integrity of HIO monolayers was determined with trans-epithelial electrical resistance (TEER) as measured with an epithelial Volt/Ohm meter (Millipore MERS 000–01). *Trypan-Blue Dye Exclusion Assay for Cell Survival*: To quantify cell survival, HIO monolayers were treated with 200μL 0.05% Trypsin/EDTA, incubated at 37°C for 5 minutes, and suspended into a single-cell homogenate. The resulting suspension was mixed with an equal volume of Trypan Blue, loaded into a dual chamber counting slide (BioRad 145–0011) and quantified using a TC20 Automated Cell Counter (BioRad 1450103).

## Histology, immunofluorescence staining, and imaging

To preserve the mucus layer and maintain bacterial attachment, HIO monolayers were fixed in Carnoy's fixative (Election Microscopy Science 64130–50) at room temperature for 4 hours, embedded in paraffin, sectioned, and stained with Hematoxylin and Eosin. Slides were also stained with Alcian Blue to identify the mucus layer and discriminate goblet cells.

## Bacterial fluorescence in situ hybridization

The tissue was fixed in Carnoy's fixative at room temperature for 4 hours, then embedded in paraffin wax. 4um sections were mounted on glass slides, baked at 60˚C for 1 hour, then de-paraffinized with xylene and dehydrated in series from 50% to 100% ethanol. The probe used was a previously validated 5' Alexafluor488-labelled universal bacterial probe (Uni519; 5'-GTATTACCGCGGCTGCTG-3') targeted to 16S rRNA. Samples were counterstained with DAPI. The probe was hybridized to the samples by adding 15μL of the 2μM probe to each slide and placing in a 45˚C hybridization chamber for 45 minutes. Slides were imaged using a Nikon Eclipse 3000 microscope.

## Gene expression profiling

The Qiagen RNeasy Mini-Kit was used to extract RNA from HIO monolayers. This was followed by gDNA elimination and cDNA conversion via the Qiagen RT2 First Strand Kit (330401). Gene expression was evaluated using the RT2 Profiler PCR Arrays for human tight junctions and human antibacterial response (330231) using the associated RT2 SYBR Green ROX qPCR Master mix (330523). Twelve samples (3 patient lineages, 4 experimental conditions), each from 2–3 pooled HIO monolayers, were run in duplicate on the ABI ViiA 7. Cycle Cut-off was established at a CT of 37 for inclusion in the analysis. All expression data was normalized to the 5 housekeeping genes included on the array and all treatment groups were normalized to the cell-line specific IC on each RT2 plate. Each RT2 plate ran all four treatment groups within a single HIO lineage to eliminate variability between runs. All samples passed all quality control metrics for PCR array reproducibility, RT efficiency, and genomic DNA contamination. All gene lists were stripped of type C errors (if 1/12 samples fell below CT cut-off), and genes with >2 difference in fold expression between replicates of any sample were excluded. Of the 169 genes analyzed, 107 were retained in the final analysis. A 1.5 difference in fold change in expression was the minimum threshold for "differential" regulation.

## Ethics statement

Human organoid cultures were obtained from existing coded samples that were originally produced and propagated by the Texas Medical Center Digestive Disease Center (TMC DDC) Organoid core from human tissue or fluids. These cultures were previously established by the TMC DDC core, and all samples were collected originally under a BCM institutional IRB-approved protocol. All organoids were established or obtained by the TMC DDC core in such a manner that the lines and samples were de-identified and the investigators on this manuscript did not have access to any existing individually identifiable private information including names, social security numbers, medical record numbers, pathology accession numbers, or any other codes that permit the samples to be linked to a living individual or associated medical information.

## Supporting information

**S1 File. Detailed protocol for Intestinal Organoid Physoxic Coculture (IOPC) system assembly.**
(DOCX)

**S2 File.** S1 Fig. Intestinal Organoid Physoxic Culture (IOPC) experimental design and modeling. S2 Fig. Measurement of transepithelial electrical resistance under physoxia. S3 Fig. Heat map comparing changes in gene expression of Je HIOs due to physoxia. S4 Fig. Schematic representation of the workflow for microbe HIO co-culture under physoxia. S5 Fig. Changes

to gene expression in different Je HIO lines in response to *B.thetaiotaomicron* co-culture. S6 Fig. Effect of co-culturing *B.thetaiotaomicron* on physoxia upregulated genes.
(DOCX)

**S1 Table. The top gene ontology pathways induced in response to physiological hypoxia.**
(XLSX)

**S2 Table. The top gene ontology pathways induced in response to *B.thetaiotaomicron* co-culture under physiological hypoxia.**
(XLSX)

## Author Contributions

**Conceptualization:** Tatiana Y. Fofanova, Joseph F. Petrosino.

**Data curation:** Tatiana Y. Fofanova.

**Funding acquisition:** Mary K. Estes.

**Investigation:** Tatiana Y. Fofanova, Jennifer M. Auchtung, Reid L. Wilson, Antonio J. Valentin, Christopher J. Stewart.

**Methodology:** Jennifer M. Auchtung, Reid L. Wilson.

**Project administration:** Tatiana Y. Fofanova, Kristi Hoffman.

**Resources:** Robert A. Britton, K. Jane Grande-Allen, Mary K. Estes, Joseph F. Petrosino.

**Software:** Tatiana Y. Fofanova.

**Writing – original draft:** Tatiana Y. Fofanova, Umesh C. Karandikar, Jennifer M. Auchtung, Christopher J. Stewart.

**Writing – review & editing:** Umesh C. Karandikar, Jennifer M. Auchtung, Mary K. Estes, Sashirekha Ramani, Christopher J. Stewart.

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
