## [Decision Letter · Decision Letter 0]

20 Nov 2023

PONE-D-23-36187A novel system to culture human intestinal organoids under physiological oxygen content to study microbial-host interaction.PLOS ONE

Dear Dr. Karandikar,

Thank you for submitting your manuscript to PLOS ONE. After careful consideration, we feel that it has merit but does not fully meet PLOS ONE’s publication criteria as it currently stands. Therefore, we invite you to submit a revised version of the manuscript that addresses the points raised during the review process.

**This study has been judged to be of interest pending some alterations that the Authors must perform.**

**They must alter the manuscript as the referees have specified in their comments and answer to all the point that have been highlighted.**

**All the amendments in the revised paper must be written in red **
**color.**

**A point to point answer rebuttal letter must be also written in details.**

We look forward to receiving your revised manuscript.

Kind regards,

Gianpaolo Papaccio, M.D., Ph.D.

Academic Editor

PLOS ONE

Journal Requirements:

4. Thank you for stating the following financial disclosure: "U19-AI116497 Mary K Estes

P30 DK-56338 H. EL-Sarag

T32GM088129, and F30 DK-108541 Wilson Reid"

5. Thank you for stating the following in the Acknowledgments Section of your manuscript: "This work was supported in part by National Institutes of Health grants U19-AI116497 (M. Estes), P30 DK-56338 (H. El-Serag), which supports the Texas Medical Centre Digestive Diseases Centre, T32GM088129, and F30 DK-108541 (R. Wilson). These funding bodies played no role in the design of the study, the collection, analysis, interpretation of data, or writing of the manuscript."

Please remove any funding-related text from the manuscript and let us know how you would like to update your Funding Statement. Currently, your Funding Statement reads as follows: "U19-AI116497 Mary K Estes

P30 DK-56338 H. EL-Sarag

T32GM088129, and F30 DK-108541 Wilson Reid".

6. Thank you for stating the following in your Competing Interests section: "No authors have competing interests".

7. In your Data Availability statement, you have not specified where the minimal data set underlying the results described in your manuscript can be found. PLOS defines a study's minimal data set as the underlying data used to reach the conclusions drawn in the manuscript and any additional data required to replicate the reported study findings in their entirety. All PLOS journals require that the minimal data set be made fully available. For more information about our data policy, please see http://journals.plos.org/plosone/s/data-availability.

8. PLOS requires an ORCID iD for the corresponding author in Editorial Manager on papers submitted after December 6th, 2016. Please ensure that you have an ORCID iD and that it is validated in Editorial Manager. To do this, go to ‘Update my Information’ (in the upper left-hand corner of the main menu), and click on the Fetch/Validate link next to the ORCID field. This will take you to the ORCID site and allow you to create a new iD or authenticate a pre-existing iD in Editorial Manager. Please see the following video for instructions on linking an ORCID iD to your Editorial Manager account: https://www.youtube.com/watch?v=_xcclfuvtxQ

9. Please include your full ethics statement in the ‘Methods’ section of your manuscript file. In your statement, please include the full name of the IRB or ethics committee who approved or waived your study, as well as whether or not you obtained informed written or verbal consent. If consent was waived for your study, please include this information in your statement as well.

10. Please include captions for your Supporting Information files at the end of your manuscript, and update any in-text citations to match accordingly. Please see our Supporting Information guidelines for more information: http://journals.plos.org/plosone/s/supporting-information.

Reviewers' comments:

Reviewer's Responses to Questions

**Comments to the Author**

1. Is the manuscript technically sound, and do the data support the conclusions?

Reviewer #1: Yes

Reviewer #2: Yes

2. Has the statistical analysis been performed appropriately and rigorously? 

Reviewer #1: Yes

Reviewer #2: Yes

3. Have the authors made all data underlying the findings in their manuscript fully available?

Reviewer #1: Yes

Reviewer #2: Yes

4. Is the manuscript presented in an intelligible fashion and written in standard English?

Reviewer #1: Yes

Reviewer #2: Yes

5. Review Comments to the Author

Reviewer #1: This study presents a new protocol do develop intestinal organoids co-culture with anearobic commensal bacteria. The study is well-conducted. However a relevant reference to a similar topic is missing (Hill DR, Huang S, Nagy MS, Yadagiri VK, Fields C, Mukherjee D, Bons B, Dedhia PH, Chin AM, Tsai YH, Thodla S, Schmidt TM, Walk S, Young VB, Spence JR. Bacterial colonization stimulates a complex physiological response in the immature human intestinal epithelium. Elife. 2017 Nov 7;6:e29132. doi: 10.7554/eLife.29132), and an analysis of transcriptional changes in terms of cell subpopulations induced after the contact with bacteria is missing (i.e. a ssGSEA on intestine cell subpopulations) in order to better understand to the question of transcriptional changes induced.

Reviewer #2: In this paper Authors generated Intestinal Organoid Physoxic Coculture (IOPC) system, and, using commensal anaerobes with varying degrees of oxygen tolerance, such as nano-aerobe Bacteroides thetaiotaomicron and strict anaerobe Blautia sp., They demonstrated that IOPC can successfully support 24 - 48 hours HIO-microbe coculture.

The paper is interesting and the experiments are well conducted.

Only little concerns must be addressed.

In figure 3c Authors must add a legend in the IF pictures referring to the markers with relative color.

Authors must add some WB to confirm the gene expression in figure 4.

However relevant references to analysis of transcriptional changes is missing (read and add Journal of Experimental and Clinical Cancer Research, 2023, 42(1), 8 “Proteotranscriptomic analysis of advanced colorectal cancer patient derived organoids for drug sensitivity prediction”; Cell Stem Cell 2020 Jan 2;26(1):17-26.e6. doi: 10.1016/j.stem.2019.10.010. Epub 2019 Nov 21. J Exp Clin Cancer Res 2023 Oct 25;42(1):281. doi: 10.1186/s13046-023-02853-4.

6. PLOS authors have the option to publish the peer review history of their article (what does this mean?). If published, this will include your full peer review and any attached files.

Reviewer #1: No

Reviewer #2: No

---

## [Author Response · Author response to Decision Letter 0]

16 Feb 2024

We thank the reviewers for their positive feedback and thoughtful suggestions. We have revised the manuscript to address the concerns raised by the reviewers. Please see our detailed responses below.

1) As suggested by Reviewer 1, we have added the reference ‘Bacterial colonization stimulates a complex physiological response in the immature human intestinal epithelium. Elife. 2017 Nov 7;6:e29132. doi: 10.7554/eLife.29132’ (line 304). 

2) Reviewer 1 suggested, “analysis of transcriptional changes in terms of cell subpopulations induced after the contact with bacteria is missing.” We believe such an analysis will need significant optimization and goes beyond the scope of this ‘proof of principle’ manuscript describing a straightforward bacterial-organoid coculture method using off-the-shelf parts. 

3) We have modified the legend to IF images in Figure 3c, as suggested by Reviewer 2. 

4) Reviewer 2 suggested, “… add some WB to confirm the gene expression in Figure 4”. 

Figure 4 shows bacteria-induced differential gene expression in the organoids. Per this suggestion, we conducted western blot analysis for Casp1 and F11R. A widely used Anti-Casp1 showed weak bands at the expected molecular weight for Casp1 and additional bands that haven’t been documented before. On the other hand, a widely used anti-F11R Ab detected multiple bands that have not been reported in previous studies. We believe the discrepancy between our results and previously published studies may reflect differences in the glycosylation pattern between cell lines and intestinal organoids. Based on these results, we conclude that the investigation of gene expression based on western blot analysis needs more significant optimization, including screening of antibodies. Such an analysis is clearly outside of the scope of this manuscript. We strongly believe that the absence of this data does not alter or diminish the coculture method described in this manuscript. 

5) Reviewer 2 listed references to be added while discussing the transcriptional changes in organoids. However, we found a reference better suited to discussing transcriptional changes in organoids during coculture with bacteria, so we added that reference (lines 304, 307). The references listed by Reviewer 2 were more appropriate to support the rationale of organoids as a preclinical model (line 350). We appreciate reviewer 2 for pointing out the need to cite relevant references while discussing transcriptional changes during bacterial coculture.

In summary, we firmly believe that the manuscript adequately describes a cost-effective and robust method to coculture intestinal organoids with anaerobic bacteria using ‘off-the-shelf parts’ that will be of immense value to the study of many researchers investigating intestinal microbes.

---

## [Decision Letter · Decision Letter 1]

4 Mar 2024

A novel system to culture human intestinal organoids under physiological oxygen content to study microbial-host interaction.

PONE-D-23-36187R1

Dear Dr. Karandikar,

We’re pleased to inform you that your manuscript has been judged scientifically suitable for publication and will be formally accepted for publication once it meets all outstanding technical requirements.

Kind regards,

Gianpaolo Papaccio, M.D., Ph.D.

Academic Editor

PLOS ONE

Additional Editor Comments (optional):

Reviewers' comments:

Reviewer's Responses to Questions

**Comments to the Author**

1. If the authors have adequately addressed your comments raised in a previous round of review and you feel that this manuscript is now acceptable for publication, you may indicate that here to bypass the “Comments to the Author” section, enter your conflict of interest statement in the “Confidential to Editor” section, and submit your "Accept" recommendation.

Reviewer #1: All comments have been addressed

Reviewer #2: All comments have been addressed

2. Is the manuscript technically sound, and do the data support the conclusions?

Reviewer #1: Yes

Reviewer #2: (No Response)

3. Has the statistical analysis been performed appropriately and rigorously? 

Reviewer #1: Yes

Reviewer #2: (No Response)

4. Have the authors made all data underlying the findings in their manuscript fully available?

Reviewer #1: Yes

Reviewer #2: (No Response)

5. Is the manuscript presented in an intelligible fashion and written in standard English?

Reviewer #1: Yes

Reviewer #2: (No Response)

6. Review Comments to the Author

Reviewer #1: Authors have addressed all reviewer's comments and concerns. No further comments.

Reviewer #2: (No Response)

7. PLOS authors have the option to publish the peer review history of their article (what does this mean?). If published, this will include your full peer review and any attached files.

Reviewer #1: No

Reviewer #2: No

---

## [Editor Report · Acceptance letter]

16 Jul 2024

PONE-D-23-36187R1 

PLOS ONE

Dear Dr. Karandikar, 

I'm pleased to inform you that your manuscript has been deemed suitable for publication in PLOS ONE. Congratulations! Your manuscript is now being handed over to our production team.

Kind regards, 

on behalf of

Prof. Gianpaolo Papaccio 

Academic Editor

PLOS ONE